# Relationship between Functions, Drivers, Barriers, and Strategies of Building Information Modelling (BIM) and Sustainable Construction Criteria: Indonesia Construction Industry

**Cakraningrat Kencana Murti**  **and Fadhilah Muslim \***

Faculty of Engineering, Universitas Indonesia, Pondok Cina, Beji, Depok City 16424, Indonesia;
cakraningrat.kencana@ui.ac.id
\* Correspondence: fadhilahmuslim@ui.ac.id

**Abstract:** With increasing sustainability concerns, such as the construction sector being responsible for using 42% of the world's energy, 30% of its raw materials, and 25% of its fresh water, building projects have been encouraged to adopt green and sustainable construction strategies. Innovation in science and technology plays an important role to support the transition to sustainable development. Its ability to rely on advanced technology and effective construction processes makes Building Information Modelling (BIM) an opportunity that can bring great benefits to the sustainable construction sector. This research focuses on functions, barriers, drivers, and implementation strategies, which were analyzed for their relationship with sustainable construction criteria using structural equation modelling (SEM). It was found that the BIM function has a positive influence on sustainable construction with relevant indicators in the form of building digitization, improvement from 2D CAD methods, and integration between tools. Relevant barriers consist of lack of demand from clients and implementation that feels like additional work. Relevant drivers consist of increasing work productivity and reducing work errors. Meanwhile, relevant strategies consist of conducting further research, providing commitment, and setting up infrastructure for the application of BIM into sustainable construction.

**Keywords:** building information modelling; function; barrier; driver; strategy; sustainable construction; Indonesia

## 1. Introduction

### 1.1. Background

As a concept, sustainability development is gaining significant attention from both the public and experts. The concept of sustainable development was first widely articulated through a publication entitled "*Our Common Future*" in 1987 issued by the World Commission on Environment and Development, also known as the Brundtland Commission, a sub-organization of the United Nations (UN) that aims to unite countries in the world in order to succeed in sustainable development [1]. Sustainable development itself does not have a standardized and binding definition, but according to the book, this concept is defined as "development that can meet the needs of the present without compromising the ability of future generations to meet their own needs." According to the book titled "*An Introduction to Sustainable Development: Routledge Perspectives on Development*" written by Jennifer A. Elliott, there are several challenges that need to be faced by society, including (a) the elimination of poverty; (b) the large gap in social inequality; and (c) the declining quality of the human environment. The idea of sustainable development is relevant in addressing these problems. This is because the concept has three main pillars that serve as guidelines in its application. The three pillars consist of social, economic, and environmental aspects [2].

The building industry is one of the many industrial sectors that contribute considerably to the three key pillars of the sustainability concept. This is reflected in the significant

outputs produced by the construction sector (buildings) in each of its life cycles, such as the fact that buildings account for one-sixth of the world's freshwater consumption, one-quarter of the world's timber harvest, and one-fifth of the world's material and energy flows [3]. In addition to the unavoidable exploitation of resources in the building industry, one of the challenges that must be addressed is that of emissions resulting from construction processes. The construction industry contributes to global warming by emitting greenhouse gases; it is anticipated that global building carbon emissions will reach 42.4 billion tons by 2035: a 43% increase over the global total carbon emissions in 2007 [4]. On the other hand, the construction industry can have positive effects on the implementation of sustainable development, such as the provision of buildings and facilities to meet the needs of human life, the provision of direct and indirect employment for the community (through other sectors that intersect with the construction industry), and its contribution to the national economy of a country. The construction industry in Australia, for instance, accounts for 7.5% of the country's gross domestic product (GDP) and generates more than one million jobs [4]. Due to the building industry's impact on sustainability, a new phrase has evolved to describe the construction industry's role in the sustainability sector in greater detail: "Sustainable Construction". In Florida, United States of America, the first international conference on sustainable construction was held in November 1994. At the conference, sustainable construction was described as "an effort to establish a healthy construction environment by employing ecologically-based, resource-efficient approaches" [5].

Today, sustainable construction is commonly characterized as the capacity to design and run a resource-conscious and healthy construction environment [6]. Considering that the building industry consumes 42% of the world's energy, 30% of its raw materials, and 25% of its clean water, resource use is vital [7]. This information demonstrates the importance of the building industry to global resource consumption, which we cannot afford to lose. Therefore, one of the primary considerations in the notion of sustainable construction is the efficiency of the construction industry [8]. In this context, sustainable construction offers a variety of alternatives, one of which is Building Information Modelling (BIM), which is presently receiving more societal attention. The word BIM has been characterized in several ways and there is no universally recognized meaning [9]. There are, however, certain broad definitions that may be utilized as recommendations for comprehending the meaning of BIM. Autodesk defines BIM as "an intelligent 3D model-based approach that delivers insights and tools for architects, engineers, and construction professionals to plan, design, construct, and manage buildings and infrastructure more effectively" [10].

BIM is not just a technology, but also a collaborative method that may provide several benefits that can be used to enhance the quality of a project [11]. The potential benefits of implementing BIM are consistent with the efficiency-related concerns previously stated, with some of the benefits of BIM adoption including, but not limited to, the following:

- Process efficiency: the capacity of BIM to integrate all parties engaged in a project in order to facilitate information sharing and decision-making throughout the project life cycle [12];
- Communication effectiveness: BIM's capacity to provide a simpler communication system and flow between parties [13];
- Efficiency in monitoring project progress: the capacity of BIM to allow direct visual monitoring of what has been completed and what remains to be completed [12];
- Improved construction planning: BIM simplifies the planning stage of a project's life cycle due to the concept of visualization of the project's activities and execution [14].

Regarding sustainability and BIM, the application of BIM to the idea of sustainability itself improves the performance of its application at each phase of a project's life cycle [15]. Due to the overlap between the features and services given by BIM and the indicators and criteria for attaining sustainability, the overall potential of utilizing and managing BIM to achieve long-term sustainability has not been realized [16].

In order to apply BIM into sustainable construction processes, it is essential to understand which factors will be evaluated. According to past research, several BIM features

are deemed significant. BIM's role is the first factor that must be thoroughly examined. According to a survey, the amount of BIM technology deployment in Indonesia itself is still low at 38% [17]. In Indonesia itself, the obstacle accounting for why the application of BIM has not been widely implemented is that people believe that BIM is still not needed, with one study reporting 38% of respondents agreeing with this claim [18]. On the other hand, the findings of the study are consistent with those of Ashraf Elhendawi's study in 2020, which covers practical techniques to persuading BIM non-users to include BIM into their workflow. Many people have indicated that they have not deployed BIM because they do not see a need for BIM features [19]. However, the study discovered that the issue stems from a lack of information among construction experts about what BIM is, what it accomplishes, and how it operates. The barrier and driver factors are the next topic that has attracted a lot of attention. Several earlier investigations, including one by Abdullah Al-Yami in 2019, have corroborated this component. The study examines the barriers and drivers of BIM adoption in the Saudi construction industry. The study investigates the state of the art of BIM, as well as the barriers and drivers, and highlights the uniqueness offered by BIM technology, such as its ability to conduct quantity surveying, life cycle assessments, designs for green building monitoring projects, and so forth [20]. This is important in order to know the potential of BIM, as well as the drawbacks that may arise in the adoption of BIM, and so it is also in line with previous research, which states that describing the factors that can serve as drivers of and obstacles to the application of BIM technology can be a practical solution in dealing with problems while increasing opportunities for those who have not yet implemented it [19].

We can now discuss the approach for using BIM in sustainable construction. Construction projects may now be examined using BIM to identify their benefits and drawbacks, as well as possibilities, while taking into consideration other elements such as financial, technical, and environmental concerns. These elements are consistent with the notion of sustainable construction with its three pillars (social, environmental, and economic). As a result, BIM technology is seen to have the capacity to assist the contemporary construction industry, and its broad use has the potential to have a considerable influence on sustainability (in this sense, sustainable construction) [21]. However, among all the benefits it may give, one of the obstacles facing its implementation is determining the best strategic approach to encourage the use of BIM in sustainable construction [22]. The purpose of this study is to examine the four areas listed above: BIM functions, drivers, barriers, and strategies for incorporating BIM into the implementation of sustainable construction, which are based on regulations governing the implementation of sustainable construction in Indonesia. There is now a rule in Indonesia that governs the implementation of sustainable construction, The Minister of Public Works and Housing released Ministry of Public Works and Housing Regulation Number 9 of 2021, which may be used as a guideline for construction service providers in Indonesia when implementing sustainable construction.

### 1.2. Problem Identification

#### 1.2.1. The Urgency of Implementing Sustainable Construction

The concept of sustainable development is rapidly evolving to achieve a sustainable relationship between social, economic, and environmental systems. This is reinforced by the growing debate in recent decades about the influence of building construction on the sustainability of the environment and human life [23]. Due to increasing sustainability issues such as the construction sector being responsible for the use of 42% of energy, 30% of raw materials, and 25% of clean water worldwide [7]. The construction sector itself is frequently referred to as an "essential economic engine" for its contribution to one-tenth of the global economy [24]. It all basically comes down to creating buildings and infrastructures for the well-being of the world's citizens while improving their quality-of-life, social interaction, and general well-being. Because of these considerations, a number of building projects have been encouraged to adopt green and sustainable construction strategies,

which are gradually being recognized as useful ways to promote the development of the construction industry [25].

1.2.2. Maximizing the Potential of Building Information Modelling (BIM)

Science and technology innovation is critical to accelerating the shift to sustainable development, particularly for cleaner manufacturing and operating processes. Building Information Modelling (BIM) technology is seen as a potential way to provide significant advantages to the architectural, engineering, and construction (AEC) sector due to its capacity to depend on modern technology and efficient building procedures [26]. Building Information Modelling (BIM) is a sustainable technology that can be used to create and track digital information about a building project throughout its entire life cycle. The implementation of BIM has attracted significant interest from both academics and practitioners, owing to its capacity to provide originality in replacing traditional project delivery methods [27]. In addition, from 2014 to 2021, BIM was the most frequently used keyword when discussing the integration of Construction 4.0, Industry 4.0, and BIM for sustainable construction in the framework of the smart city [28]. Concerning the notion of sustainability, the use of BIM enables greater performance in the application of the concept of sustainability at each step of a project's life cycle [15]. However, owing to the overlap of BIM features and services with indicators and criteria for attaining sustainability, the total potential of utilizing and managing BIM has not been leveraged to accomplish sustainability over the life cycle of a building [16]. Practitioners, on the other hand, have indirectly used BIM to measure, evaluate, and support several sustainability-related indicators, given the breadth of BIM functions that allow every involved and responsible party in the project to exchange knowledge related to the ongoing project in each different dimensional model, such as three-dimensional models (3D), time-related models (4D), cost-related models (5D), energy and performance analysis models (6D), and a variety of other models [15]. To optimize this, an integrated methodology or model that permits the use of BIM in the application of each sustainability indicator for each life phase of each project is required [16].

*1.3. Research Questions*

Based on the problem identification above, the objectives of this research include:

- Analyzing the relationship between BIM functions and sustainable construction criteria;
- Analyzing the relationship between barrier factors in the application of BIM in sustainable construction and sustainable construction criteria;
- Analyzing the relationship between the driving factors for the application of BIM in sustainable construction and sustainable construction criteria;
- Analyzing the relationship between strategies for improving the application of BIM in sustainable construction and sustainable construction criteria.

*1.4. Limitations of The Research*

To ensure that this research stays within the scope intended, we restrict this study with varying conditions, including:

- The construction service players in Indonesia are the subjects for this study;
- The construction service players are drawn from institutions/organizations that have implemented or are using BIM technology and sustainable construction in their workflows;
- The construction service players are from companies/institutions that are willing and competent to abide by Indonesia's Ministry of Public Works and Public Housing Regulation number 9 of 2021;
- Structural equation modelling is used for the analysis process.
- The criteria for sustainable construction employed are based on Indonesia's Ministry of Public Works and Public Housing Regulation number 9 of 2021, which covers the standards for implementing sustainable construction in Indonesia.

*1.5. Functions of BIM for Sustainable Construction*

The construction project procedure is guided by a lengthy life cycle. It is a series of stages beginning with the initiating phase of a project and ending with the closing phase, which involve complex paperwork and information. The reality of construction projects requires the collaboration and integration of diverse specialists from each stakeholder in order to fulfill the responsibilities outlined in their respective project scopes and objectives. However, diverse types of segmented or scattered information and data among project stakeholders lead to misunderstandings, which must be clarified and addressed to prevent dissatisfaction, lack of confidence, and conflict among stakeholders. Therefore, conventional objectives (i.e., time, cost, and quality) and project productivity are frequently impacted by these issues [9]. BIM technologies are frequently used when they can reduce the complexity and difficulty of project management by addressing the volume of information, the quality of information created, and the description of this information. Thus, the construction industry tends to shift from traditional construction processes to BIM-based procedures to eliminate issues [29]. Before we can implement BIM, we must first understand its capabilities. Regarding the scope of BIM's coverage in relation to its capacity, BIM is classified into different levels of maturity, called "dimension". This dimension begins with 3D, a virtual mock-up model that, among other things, visually expresses design concepts in the three primary spatial dimensions (width, height, and depth) [30]. This level of maturity has several familiar functions in the field, such as project visualization, collision detection, and model walkthroughs [31–33]. However, despite the capabilities of 3D BIM, it is not enough; in order to achieve faster delivery, time is taken into account and comprises a fourth dimension. On the basis of recent advancements in Building Information Modelling (BIM), four-dimensional (4D) technology, time-dependent structural analysis, and collision detection, we present and establish a 4D structural information model [34]. Additionally, BIM does not end in the fourth dimension. For BIM to reach its maximum potential, numerous other dimensions must be added. Currently, there are seven dimensions for BIM including scheduling, estimating, sustainability, and facility management. This extensive range of BIM dimensions has enabled it to serve various purposes in the construction process, such as quantity take-off, life cycle analysis, risk management, asset management, quality management, and so forth [15,20,29,34]. Yet, despite the power of BIM, the majority of AEC sectors in Indonesia are still utilizing conventional methods such as 3D modeling and CAD, and there is no law from the Indonesian government forcing the use of BIM in the building design and construction process, as most regulations are advisory only [35].

*1.6. Barriers to BIM Implementation*

A barrier is something, such as a rule, law, or policy, that makes it difficult or impossible for something to occur or be attained. A barrier can also be defined as a factor that prevents two individuals or groups from agreeing, communicating, or cooperating. Despite the functions and benefits of the adoption of BIM technology, its implementation has been limited thus far due to several barriers. A study discovered that BIM awareness, knowledge, and interest vary across construction industry disciplines, but perceptions of the main factors affecting its implementation are consistent among engineers, architects, project managers, and other key stakeholders. They tend to categorize socio-organizational barriers (e.g., resistance to change); financial barriers (e.g., cost of BIM training, software, and hardware); technical barriers (e.g., interoperability issues); contractual barriers (e.g., lack of BIM-related aspects in current contracts); and legal barriers (e.g., ownership of BIM models, intellectual property, and copyright issues) as the five main classifications of barriers to BIM adoption [9]. For the study case in this research, Indonesia, research conducted by Sriyolja et al. in 2021 mentioned that it is possible to adopt BIM in Indonesia to find better answers to these challenges. Creating regulations that serve as a guide for consultants and contractors at work is one method. This is due to the fact that Indonesia is a developing country where the majority of development projects are government-owned. Occasionally, government regulations will assist in overcoming other barriers [36].

### 1.7. Drivers of BIM Implementation

A driver is an action or entity that acts as an enabler, catalyst, or motivator, intriguing someone to act or causing something to occur. Several studies have identified several advantages of and implementation drivers for BIM. Some studies evaluate the advantages of BIM as: technical excellence, interoperability, early building information capture, applicability throughout the building life cycle, integrated procurement, improved cost control mechanisms, conflict reduction, and project team advantages. Throughout the life cycle of a building, BIM increases productivity and facilitates the management of project information. In addition, BIM contributes to the increased productivity and efficiency of the construction process as well as the general improvement of project value and construction practices [29]. A survey was conducted on the world's top construction companies, and it was found that the key project-related benefits contractors receive from BIM are: reduced rework, reduced construction costs, reduced project duration, and improved safety—all of which have a strong impact on the company's return on investment. Drivers of BIM implementation include: improved collaboration on projects, clash detection, improved ability to respond to information requests, improved cost estimation and controllability, improved client satisfaction, improved product quality, improved quality of construction details, improved ability to meet sustainability needs, and facilitated cost savings during design [9].

### 1.8. BIM Implementation Strategies

Implementation strategies are needed so that BIM technology can be implemented appropriately. This is to ensure that the implementation is not done without basis and to minimize the risk of failure that can occur throughout the project. One of the challenges to integrate BIM into sustainable construction lies in considering the proper strategic directions to promote the application of BIM in sustainable developments [22]. BIM is well-suited for sustainable building projects and applications requiring data on sustainability and energy efficiency [37]; however, it can be utilized in a range of industries. It is essential to conduct additional studies to attain a deeper understanding of BIM adoption strategies for sustainable construction projects in Malaysia. Prior studies have not completely addressed these techniques; however, they must be carefully evaluated in order to design and deploy successful programs for enhancing BIM technology use in sustainable construction projects [22].

## 2. Materials and Methods

### 2.1. Research Methods

The first step in this research consisted of a literature review on the subject at hand. This rendered a determination of the problem's fundamentals easier and clarified the ways earlier research studies approached comparable situations. Based on the research questions (RQ) that were created, it was next decided how strategies could be implemented in the form of research methodologies. This study used four research questions that were drawn from the problem's context. These four research topics were posed concurrently, with the "heart" of this study being an examination of the link between BIM and sustainable construction requirements. As a depiction of BIM, four key variables were employed, including BIM function factors, barrier factors, driving factors, and BIM implementation strategies, based on the explanations provided in the previous section. Each RQ was allocated these four elements with the objective of determining the link between each of these factors and sustainable construction criteria. Due to the similarity of the four RQs, they were processed in parallel, with the first phase consisting of the generation of variables and their corresponding indicators. It is essential to understand what components reflect and explain the variables used in this study. To guarantee that the employed variables were accurate, the next stage consisted of expert validation. Validation was conducted by experts in their respective domains to see if the variables and indicators utilized were representative. Additionally, questions for respondents were assembled. Before they could be given to respondents, a pilot survey was conducted to see if the research questionnaire could be

comprehended and whether it was straightforward to read by the anticipated audience. After all variables and research questionnaires had been validated, data dissemination and collecting was carried out. Once the data were collected, they were processed using one of two methods: structural equation modelling (SEM), more specifically SEM PLS (partial least square), because the basis of this research is more oriented toward the preparation of theory, and the relative importance index (RII), which was used to determine which relevant indicators were contained within each variable. To analyze data, it was necessary to be aware of the essential indications included in each BIM variable. Using SEM, the link between each BIM variable and the phases of the sustainable construction criterion were also determined (planning, programming, and construction stages). Thereafter, we applied important indicators that have been studied using confirmatory factor analysis (CFA) and relative importance index (RII) methodologies for each link between variables, based on the stage at which each of the existing BIM variables had a strong significance. From there, we drew findings and generated recommendations for future research.

### *2.2. Research Variables*

To be able to analyze the relationship in this research, a series of variables were needed, which represented the function, barrier, driver, and strategy aspects of this research. Each variable consists of their own respective sub-variables and indicators. These components, such as variables, sub-variables, and indicators, are addressed in the following table, starting with variables and indicators for BIM function in Table 1, barriers in Table 2, drivers in Table 3, and strategies in Table 4.

These preceding tables were assigned for each exogenous construct (X). However, for this relationship model to perform as intended, these variables were not enough to explain the relationship between BIM and sustainable construction. Thus, the following Table 5 was addressed to explain more about the sustainable construction criteria, an endogenous construct (Y) for this relationship model.

**Table 1.** Variables and Indicators for BIM Functions.

| Code | Variables | Code | Indicators | References |
|------|-----------|------|-----------|-----------|
| X1.1 | Design Phase | X1.1.1 | Building digitization | [38,39] |
| | | X1.1.2 | Integration between parties involved | [40,41] |
| | | X1.1.3 | Clash detection | [38–41] |
| | | X1.1.4 | Feasibility study | [41,42] |
| | | X1.1.5 | Energy efficiency analysis | [39–41] |
| X1.2 | Build Phase | X1.2.1 | Collaboration platform for stakeholders | [41,42] |
| | | X1.2.2 | Improve realization based on standards | [38–42] |
| | | X1.2.3 | Improve understanding of original design | [39,42] |
| | | X1.2.4 | Further development of 2D CAD | [40] |
| | | X1.2.5 | Work Efficiency | [39] |
| X1.3 | Operate Phase | X1.3.1 | Minimization of document errors | [41,42] |
| | | X1.3.2 | Become the main source of data | [42] |
| | | X1.3.3 | Become a database of asset information | [38–42] |
| | | X1.3.4 | Integration between different tools for each party | [38–42] |
| | | X1.3.5 | Asset performance analysis | [38–42] |

### *2.3. Respondent Criteria*

#### 2.3.1. Respondent Criteria for Expert Validation

Data collection for expert validation by three experts was carried out as part of the preparation of a sustainable BIM model. The purpose of expert validation was to evaluate whether the variables proposed previously in this study were not suitable or still required additions. The data collection itself was carried out using a questionnaire survey instrument with the following expert criteria:

- For experts from the practitioner field, the minimum required education level is Bachelor/Equivalent, a minimum required work experience of at least 15 years in the construction field, and experience implementing sustainable construction or BIM in past workflows;
- For experts from academia, the minimum required education level is Bachelor/ Equivalent with at least 15 years of teaching experience and an understanding of sustainable construction or BIM.

**Table 2.** Variables and Indicators for Barrier Factors.

| Code | Variables | Code | Indicators | References |
|---|---|---|---|---|
| X2.1 | Sustainable Construction Barrier | X2.1.1 | Lack of training and education related to construction sustainable | [22,43–45] |
| | | X2.1.2 | Cost of implementation construction that tend to be high | |
| | | X2.1.3 | Lack of experts in sustainable construction | |
| | | X2.1.4 | Lack of demand from clients | |
| | | X2.1.5 | Tendency not to adapt into sustainable construction | |
| X2.2 | Building Information Modelling Barrier | X2.2.1 | BIM implementation temporarily reduces work efficiency due to the unwillingness of related parties to adapt | [22,43–45] |
| | | X2.2.2 | BIM implementation requires higher initial investment than conventional methods | |
| | | X2.2.3 | BIM implementation not required by client | |
| | | X2.2.4 | BIM implementation causes delays in the project due to lack of experience in the use of BIM by related parties | |
| | | X2.2.5 | BIM implementation feels like additional work that must be done | |

**Table 3.** Variables and Indicators for Driver Factors.

| Code | Variables | Code | Indicators | References |
|---|---|---|---|---|
| X3.1 | Sustainable Construction Driver | X3.1.1 | Improve energy efficiency | [43–47] |
| | | X3.1.2 | Improve resource conservation | |
| | | X3.1.3 | Improve indoor environmental quality | |
| | | X3.1.4 | Reduce construction waste | |
| | | X3.1.5 | Reduce negative impact on the environment | |
| | | X3.1.6 | Improve productivity | |
| | | X3.1.7 | Reduce operation and maintenance costs after construction | |
| | | X3.1.8 | Increase job opportunity | |
| | | X3.1.9 | Improve health and safety | |
| | | X3.1.10 | Improve Workers' Welfare | |
| X3.2 | Building Information Modelling Driver | X3.2.1 | Create a more effective design process | [43–47] |
| | | X3.2.2 | Reduce error and risk | |
| | | X3.2.3 | Provide building life cycle data for operations and maintenance | |
| | | X3.2.4 | Provide lessons learned from previous projects | |
| | | X3.2.5 | Provide a basis for justifying a design based on standards and regulations | |
| | | X3.2.6 | Centralize information in the database | |
| | | X3.2.7 | Improve overall project cost efficiency | |
| | | X3.2.8 | Improve visualization for stakeholders | |
| | | X3.2.9 | Create more controlled project scheduling | |
| | | X3.2.10 | Improved work safety on projects | |

**Table 4.** Variables and Indicators for Implementation Strategies.

| Code | Variables | Code | Indicators | References |
|---|---|---|---|---|
| X4.1 | Awareness Based | X4.1.1 | Organize workshops, trainings, and events to increase awareness | [22,25,48,49] |
| | | X4.1.2 | Improve utilization of information technology | |
| | | X4.1.3 | Conduct more research in the use of BIM for sustainable construction | |
| | | X4.1.4 | Provide comprehensive support from every area | |
| X4.2 | Regulation Based | X4.2.1 | More detailed standards and regulations in the use of BIM for sustainable construction | [22,25,48,49] |
| | | X4.2.2 | Create encouragement, support, and commitment from regulators to accommodate the implementation of BIM for sustainable construction | |
| | | X4.2.3 | Development of operational standards and procedures in the application of BIM for sustainable construction | |
| X4.3 | Company Based | X4.3.1 | Better information availability in the integration of BIM into sustainable construction | [22,25,48,49] |
| | | X4.3.2 | Prepare infrastructure (tools, software, hardware, etc.) to be able to apply BIM into sustainable construction | |
| | | X4.3.3 | Recruit experts and specialists in BIM | |
| | | X4.3.4 | Recruit experts and specialists in Sustainable Construction | |
| | | X4.3.5 | Develop a dedicated team on the implementation of BIM into sustainable construction | |

### 2.3.2. Respondent Criteria for Pilot Survey

After the existing variables were validated by experts and considered valid, the next step was to conduct a pilot survey. The pilot survey itself was intended to determine the understanding of prospective questionnaire respondents, which was represented by 10 respondents. Respondents of this pilot survey were faced with questions that were later used in the questionnaire survey, and they were asked to assess the questions and provide input regarding whether the questions used were easy to understand or not. The criteria of the pilot survey were:

- Respondents must have a Bachelor's degree or equivalent at minimum;
- Respondents must have at least 1 year of working experience in the construction field;
- Respondents must have implemented or must be currently implementing sustainable construction or BIM in their workflows.

**Table 5.** Variables and Indicators for Sustainable Construction Criteria [50].

| Code | Variables | Code | Indicators |
|---|---|---|---|
| Y1 | Initiating phase | Y1.1.1 | The suitability of the location of the development plan in accordance with the standards |
| | | Y1.2.1 | Land suitability with function based on area master plan |
| | | Y1.3.1 | Availability of disaster risk mitigation plan |
| | | Y1.4.1 | Availability of local construction resource utilization plan |
| | | Y1.5.1 | Availability of development plans that are responsive to gender, people with disabilities, and marginalized people |
| | | Y1.6.1 | Availability of development plans that support regional economic development |
| | | Y1.7.1 | Availability of development plans in accordance with technical standards and utilization of environmentally friendly technology |

**Table 5.** *Cont.*

| Code | Variables | Code | Indicators |
|---|---|---|---|
| Y2 | Planning Phase | Y2.1.1 | Availability of a community accessibility plan as part of the economic viability of sustainable construction |
| | | Y2.2.1 | Availability of detailed engineering design (DED) of sustainable construction buildings |
| | | Y2.2.2 | Availability of land for sustainable construction buildings |
| | | Y2.2.3 | Availability of environmental approvals |
| | | Y2.3.1 | Availability of feasibility study documents |
| | | Y2.4.1 | Provide responses to community aspirations |
| | | Y2.5.1 | Design compliance with principles that are responsive to gender, people with disabilities, and marginalized people |
| | | Y2.6.1 | Availability of natural resource utilization efficiency programs |
| | | Y2.7.1 | Availability of building technical requirements and criteria |
| Y3 | Construction phase | Y3.1.1 | Availability of health and safety management system |
| | | Y3.1.2 | The use of lightning rods according to standards |
| | | Y3.2.1 | Provide land use efficiency and minimize changes in land condition |
| | | Y3.3.1 | Provide energy conservation and efficiency |
| | | Y3.4.1 | Provide water utilization efficiency |
| | | Y3.4.2 | Construction of water catchment area |
| | | Y3.5.1 | The use of environmentally friendly construction materials |
| | | Y3.5.2 | The use of local construction materials |
| | | Y3.5.3 | Provide construction material use efficiency |
| | | Y3.5.4 | The use of recycled construction materials |
| | | Y3.5.5 | The use of prefabricated construction materials |
| | | Y3.6.1 | Maintain air water quality |
| | | Y3.6.2 | Provide noise reduction |
| | | Y3.7.1 | Provide solid and liquid waste management |
| | | Y3.7.2 | Construction of building drainage system |
| | | Y3.7.3 | Provide disaster adaptation |
| | | Y3.8.1 | Community involvement |
| | | Y3.9.1 | Construction of ender, disability, and marginalized responsive facility |
| | | Y3.10.1 | Construction of community access and interaction spaces |
| | | Y3.10.2 | Construction of public transport user access and facilities |
| | | Y3.10.3 | Construction of pedestrian and/or cycling access and facilities |
| | | Y3.11.1 | Design compliance with building construction technical requirements and criteria |

2.3.3. Respondent Criteria for Research Questions

The respondent survey was used to collect the required data related to this research from sources that are in line with this research. For the number of respondents themselves, we referred to Wynne W. Chin in his article entitled "Partial Least Squares for IS Researchers: An Overview and Presentation of Recent Advances Using The PLS Approach" in 2000, where the number of respondents for SEM PLS-based research ranges from 30 to 100 respondents [51], with the following criteria:

- Respondents must have Bachelor's degree or equivalent at minimum;
- Respondents must have at least 1 year of working experience in the construction field;
- Respondents must have implemented or must currently be implementing sustainable construction or BIM in their workflows.

*2.4. Research Analysis Method*

2.4.1. Data Analysis for Expert Validation

The first phase of data analysis consisted of identifying and evaluating the indicators utilized in the creation of the sustainable BIM model. The various metrics that reflect BIM and sustainability factors were analyzed using current literature. Expert responders then reevaluated these factors to see whether any variables did not fit or had not been included.

The validation process itself was undertaken via a questionnaire or interview. The variables were rearranged based on the inputs collected during the expert validation phase or not at all if there were no inputs. Theoretically, expert validation is an instrument validation procedure conducted via expert review or justification or through the evaluation of a set of panels consisting of individuals who are experts in the topic or content of the variables undergoing assessment. Our expert validation employed the Delphi Method: a technique for surveying and collecting the opinions of experts on certain themes [52].

### 2.4.2. Data Analysis for Pilot Survey

A pilot survey is a small-scale version of a larger survey used to evaluate the feasibility and validity of a survey's concept and methodology [53]. Prior to conducting the larger survey, the primary objective of a pilot survey is to discover any concerns or problems that may develop during the real survey, such as confusing questions, difficulty in gathering replies, and misunderstandings of data. A sample of the target population is selected and requested to complete a survey in a pilot study. The results gathered from the pilot survey are then reviewed to assess whether the questions are clear and simple to comprehend, whether the survey technique is acceptable and successful, and whether the collected data are useful and relevant to the study goals. Before the real survey is done, the data from the pilot survey may be utilized to make any required revisions to the survey's design, questionnaire, and methodology. For this study, the sample size was represented by 10 respondents. Respondents of the pilot survey were presented with the questions that would later be used in the questionnaire survey and they were asked to rate and provide feedback on whether the questions were easy to understand or not.

### 2.4.3. Data Analysis for Structural Equation Modelling (SEM)

The analysis carried out at this stage was conducted using Structural Equation Modelling (SEM). SEM is divided into two kinds, namely covariance-based SEM (CB-SEM) and partial least square SEM (SEM PLS). Covariance-based SEM generally tests causality or theory while SEM PLS is more directed towards predictive models. However, there is a difference between covariance-based SEM and component-based SEM PLS. Namely, they differ in their respective utilizations of structural equation models to test theories or theory developments that aim to make predictions [54]. In this study, the SEM approach used was SEM PLS, considering that this research is more predictive and helpful for building a new theory, rather than for testing an existing theory. There are several software programs that can be used in analyzing SEM, but in this study, we used the SMARTPLS 4® application. Before processing could be done using the SEM method, first, the data obtained from the respondent survey needed to be tested. Data testing consisted of two stages, namely testing for the measurement model and for the structural model as we can see from Figure 1.

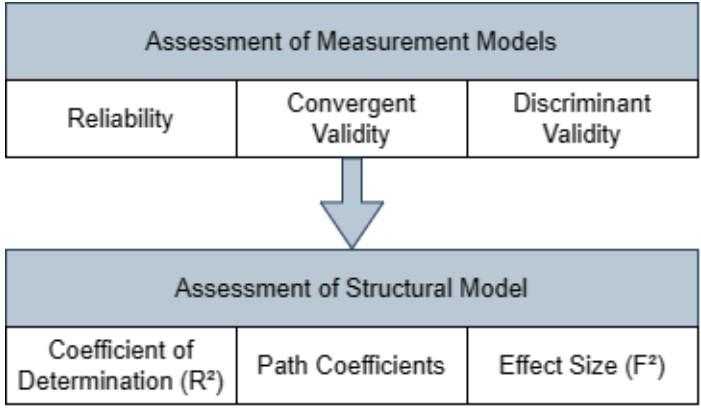

**Figure 1.** Testing and data analysis for SEM (Reprinted/adapted with permission from Ref. [55]. 2019. Solla et al.).

Measurement models were evaluated using algorithms in statistical applications to determine reliability and construct validity as well as discriminant validity and loading of all construct indicators [55]. A model was considered reliable when the composite reliability and Cronbach's alpha of each construct was equal to or greater than 0.70 [56]. The next step was structural model testing, which was developed after the measurement model had been validated. The structural model was assessed using algorithms in the SEM data processing application. The basic concepts for assessing the structural model included:

- Coefficient of determination ($R^2$)
  The coefficient of determination ($R^2$) describes the degree of explained variance of the dependent latent variable. It is used to determine the explanatory power of the structural model [57]. $R^2$ must be met, where values between 0.02 and 0.12 are considered weak, values between 0.13 and 0.25 are considered moderate, and values of 0.26 and greater are considered substantial;

- Path coefficient
  The second criterion for SEM evaluation involves assessing the path coefficient; it measures the strength of the relationship between the latent variables of the research model, where the significant value should be at least 0.05 [58];

- Effect size ($F^2$)
  The $F^2$ effect size is a measure of the impact of a particular predictor construct on an endogenous construct. In addition to assessing the size of the $R^2$ values of all endogenous constructs, the $F^2$ effect size can also be calculated for each construct. An $F^2$ effect size of 0.02 is considered small; 0.15, moderate; and 0.35, strong [57].

After all stages of testing, with both measurement model and structural model testing having been completed, the data that were collected and validated were processed into SEM form with the help of SEM data processing software.

### 2.4.4. Data Analysis for Relative Importance Index (RII)

The data analysis process was carried out with a relative importance index (RII) approach to determine what were the most relevant indicators of each BIM variable for functions, barriers, drivers, and strategies. Relative Importance Index (RII) was used to determine the relative importance of the quality factors involved [59]. The Likert scale points used were equal to the W value, which is the weight given to each factor by the respondents:

$$\text{RII} = \frac{\sum W}{A \times N},$$

(1)

where:
RII: Relative Importance Index,
W: The weighting given to each factor,
A: The highest weight present in the research data, and
N: Total respondents who filled in the data

To determine their importance, each factor was based on the higher Relative Importance Index (RII) value obtained from the equation above. Different factors had different Relative Importance Index (RII) values and these were used to rank the factors. After analysis using structural equation modelling, a sustainable BIM model was obtained that connects each BIM attribute from each RQ, such as functions, barrier factors, driver factors, and strategies, to be analyzed for its relevance and relationship with each sustainable construction criteria. The results of the analysis were then validated by experts to find out whether the matrix from the model analysis was correct and relevant.

## 3. Results

### 3.1. Respondent Profile

In this study, a respondent questionnaire was used to collect the required data related to the research. Based on predetermined criteria, it was known that the number of respon-

dents was in the range of 30 to 100 people. Questionnaires were distributed to employees in the construction sector based on predetermined criteria through the LinkedIn social media website and a total of 60 respondents filled out the questionnaire. From the 60 respondents who filled out the questionnaire, a grouping was compiled consisting of the three that we considered met the criteria mentioned above, namely regarding education, length of work experience, and the sector where they currently work. We thus generated the following profile grouping:

From the questionnaires obtained from research respondents, it was determined that 54 respondents (90%) held a Bachelor's or equivalent degree, and that 6 respondents (10%) held a Master's or equivalent degree, as we can see from Figure 2.

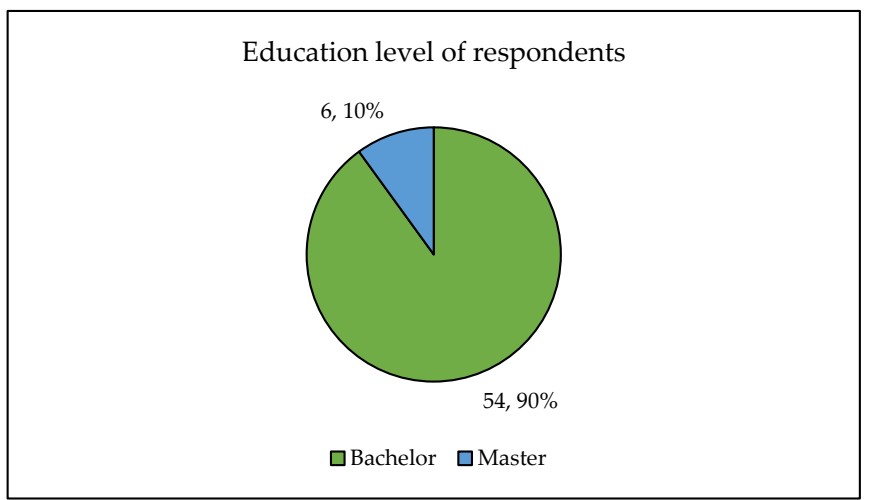

**Figure 2.** Education level of respondents.

From the questionnaires obtained from research respondents, it was determined that 12 respondents (20%) had work experience totaling 1 year, 28 respondents (47%) had work experience totaling 2–5 years, 15 respondents (25%) had work experience totaling 6–10 years, 3 respondents (5%) had work experience totaling 11–15 years, no respondents (0%) had work experience totaling 16–20 years, and 2 respondents (3%) had work experience totaling more than 20 years, as we can see from Figure 3.

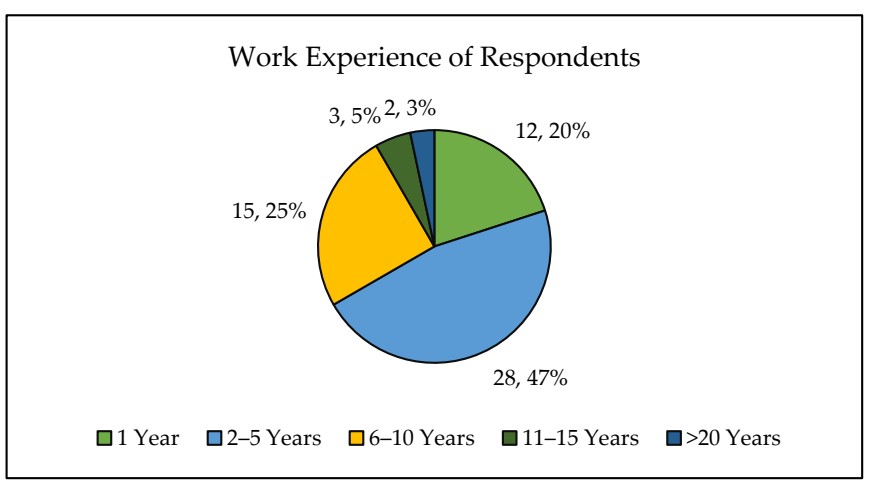

**Figure 3.** Work experience of respondents.

From the questionnaires obtained from research respondents, it was determined that 41 respondents (68%) were contractors, 2 respondents (4%) were owners, and 17 respondents (28%) were consultants, as we can see from Figure 4.

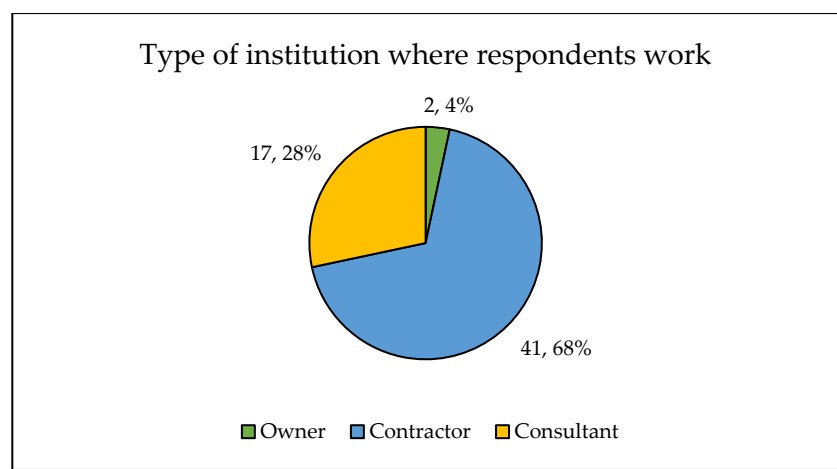

**Figure 4.** Type of institution where respondents work.

*3.2. Research Results*

3.2.1. Results of Structural Equation Modelling Testing

Testing of the four research questions (RQ) used a structural equation modelling (SEM) approach with input data from the 60 respondents who filled out the questionnaire survey, where the application used was SMARTPLS® software. Using this application, the relationship between the proposed model and the attributes in the use of BIM technology was obtained, which in this case consisted of the BIM function, barrier factors, driver factors, and strategies in the application of BIM to the criteria for implementing sustainable construction in Indonesia according to Minister of Public Works and Public Housing Regulation No. 9 of 2021. The model is shown in the following Figure 5:

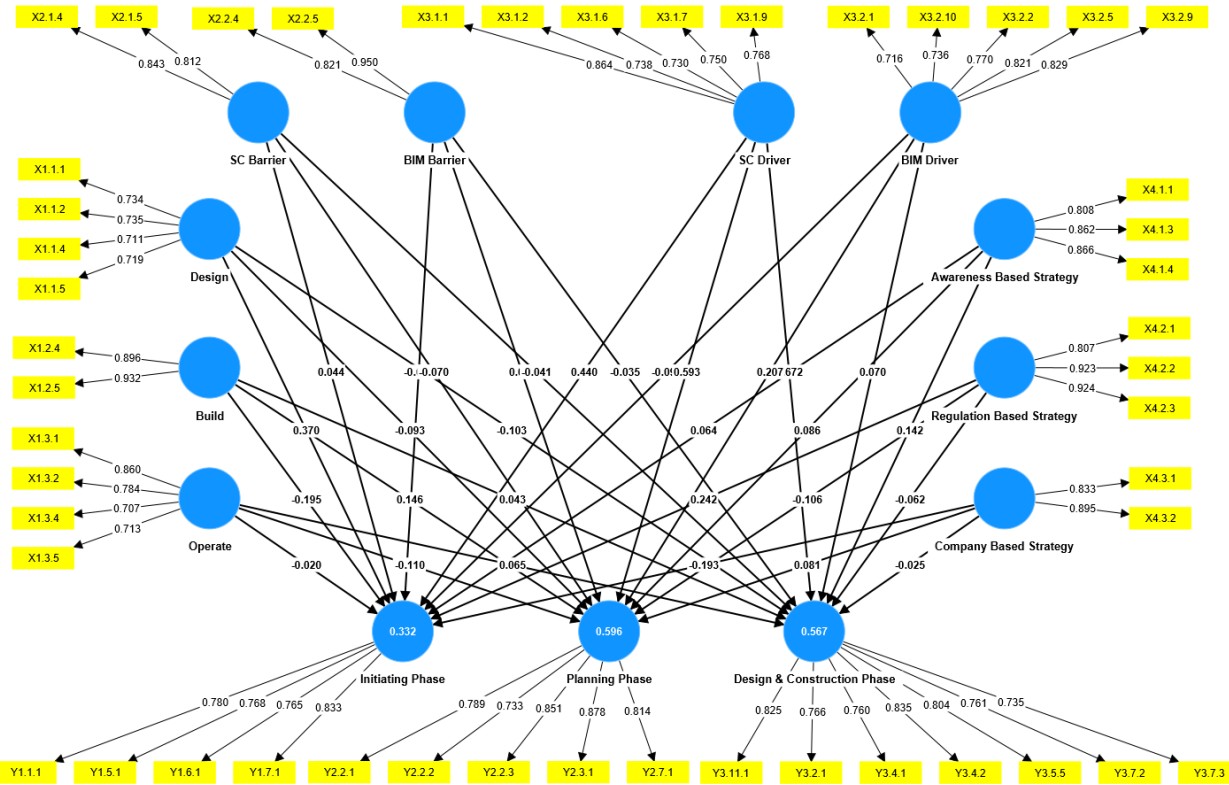

**Figure 5.** SEM model of the relationship between BIM factors and Sustainable Construction Implementation Criteria in Indonesia.

After modelling and data input, the test results were obtained to test the outer model and inner model of the proposed SEM model, with the summary results listed in the following Table 6:

**Table 6.** Summary of SEM model data testing.

| Code | Variable | Code | Indicator | Outer Loading | Average Variance Extracted | Composite Reliability | Cronbach's Alpha | R² | Q² |
|------|----------|------|-----------|---------------|----------------------------|------------------------|-------------------|-----|-----|
| X1.1 | Design Phase | X1.1.1 | Building digitization | 0.734 | 0.525 | 0.700 | 0.701 | - | - |
| | | X1.1.2 | Integration between parties involved | 0.735 | | | | | - |
| | | X1.1.4 | Feasibility Study | 0.711 | | | | | - |
| | | X1.1.5 | Energy efficiency analysis | 0.719 | | | | | - |
| X1.2 | Build Phase | X1.2.4 | Further development of 2D CAD | 0.896 | 0.837 | 0.830 | 0.807 | - | - |
| | | X1.2.5 | Work Efficiency | 0.932 | | | | | - |
| X1.3 | Operate Phase | X1.3.1 | Minimization of document errors | 0.860 | 0.590 | 0.809 | 0.781 | - | - |
| | | X1.3.2 | Become the main source of data | 0.784 | | | | | - |
| | | X1.3.4 | Integration between different tools for each party | 0.707 | | | | | - |
| | | X1.3.5 | Asset performance analysis | 0.713 | | | | | - |
| X2.1 | Sustainable Construction Barrier | X2.1.4 | Lack of demand from clients | 0.843 | 0.685 | 0.785 | 0.741 | - | - |
| | | X2.1.5 | Tendency not to adapt into sustainable construction | 0.812 | | | | | - |
| X2.2 | Building Information Modelling Barrier | X2.2.4 | BIM implementation causes delays in the project due to lack of experience in the use of BIM by related parties | 0.821 | 0.788 | 0.951 | 0.752 | - | - |
| | | X2.2.5 | BIM implementation feels like additional work that must be done | 0.950 | | | | | - |
| X3.1 | Sustainable Construction Driver | X3.1.1 | Improve energy efficiency | 0.864 | 0.595 | 0.834 | 0.83 | - | - |
| | | X3.1.2 | Improve resource conservation | 0.738 | | | | | - |
| | | X3.1.6 | Improve productivity | 0.730 | | | | | - |
| | | X3.1.7 | Reduce operation and maintenance costs after construction | 0.750 | | | | | - |
| | | X3.1.9 | Improve health and safety | 0.768 | | | | | - |
| X3.2 | Building Information Modelling Driver | X3.2.1 | Create a more effective design process | 0.716 | 0.602 | 0.901 | 0.84 | - | - |
| | | X3.2.2 | Reduce error and risk | 0.736 | | | | | - |
| | | X3.2.5 | Provide a basis for justifying a design based on standards and regulations | 0.770 | | | | | - |
| | | X3.2.9 | Create more controlled project scheduling | 0.821 | | | | | - |
| | | X3.2.10 | Improved work safety on projects | 0.829 | | | | | - |

**Table 6.** *Cont.*

| Code | Variable | Code | Indicator | Outer Loading | Average Variance Extracted | Composite Reliability | Cronbach's Alpha | R² | Q² |
|---|---|---|---|---|---|---|---|---|---|
| X4.1 | Awareness-Based | X4.1.1 | Organize workshops, trainings, and events to increase awareness | 0.808 | 0.715 | 0.801 | 0.8 | - | - |
| | | X4.1.3 | Conduct more research in the use of BIM for sustainable construction | 0.862 | | | | | - |
| | | X4.1.4 | Provide comprehensive support from every area | 0.866 | | | | | - |
| X4.2 | Regulation-Based | X4.2.1 | More detailed standards and regulations in the use of BIM for sustainable construction | 0.807 | 0.786 | 0.922 | 0.867 | - | - |
| | | X4.2.2 | Create encouragement, support, and commitment from regulators to accommodate the implementation of BIM for sustainable construction | 0.923 | | | | | - |
| | | X4.2.3 | Development of operational standards and procedures in the application of BIM for sustainable construction | 0.924 | | | | | - |
| X4.3 | Company-Based | X4.3.1 | Better information availability in the integration of BIM into sustainable construction | 0.833 | 0.747 | 0.785 | 0.764 | - | - |
| | | X4.3.2 | Prepare infrastructure (tools, software, hardware, etc.) to be able to apply BIM into sustainable construction | 0.895 | | | | | - |
| Y1 | Initiating Phase | Y1.1.1 | The suitability of the location of the development plan in accordance with the standards | 0.780 | 0.620 | 0.796 | 0.795 | 0.332 | 0.13 |
| | | Y1.5.1 | Availability of development plans that are responsive to gender, people with disabilities, and marginalized people | 0.768 | | | | | 0.074 |
| | | Y1.6.1 | Availability of development plans that support regional economic development | 0.765 | | | | | 0.016 |
| | | Y1.7.1 | Availability of development plans in accordance with technical standards and utilization of environmentally friendly technology | 0.833 | | | | | 0.105 |

**Table 6.** *Cont.*

| Code | Variable | Code | Indicator | Outer Loading | Average Variance Extracted | Composite Reliability | Cronbach's Alpha | R² | Q² |
|------|----------|------|-----------|---------------|----------------------------|-----------------------|------------------|-----|-----|
| Y2 | Planning Phase | Y2.2.1 | Availability of detailed engineering design (DED) of sustainable construction buildings | 0.789 | | | | | 0.123 |
| | | Y2.2.2 | Availability of land for sustainable construction buildings | 0.733 | 0.663 | 0.888 | 0.873 | 0.596 | 0.022 |
| | | Y2.2.3 | Availability of environmental approvals | 0.851 | | | | | 0.294 |
| | | Y2.3.1 | Availability of feasibility study documents | 0.878 | | | | | 0.234 |
| | | Y2.7.1 | Availability of building technical requirements and criteria | 0.814 | | | | | 0.298 |
| Y3 | Construction Phase | Y3.2.1 | Provide land use efficiency and minimize changes in land condition | 0.825 | | | | | 0.21 |
| | | Y3.4.1 | Provide water utilization efficiency | 0.766 | | | | | 0.129 |
| | | Y3.4.2 | Construction of water catchment area | 0.760 | | | | | 0.107 |
| | | Y3.5.5 | The use of prefabricated construction materials | 0.835 | 0.616 | 0.896 | 0.895 | 0.567 | 0.158 |
| | | Y3.7.2 | Construction of building drainage system | 0.804 | | | | | 0.172 |
| | | Y3.7.3 | Provide disaster adaptation | 0.761 | | | | | 0.209 |
| | | Y3.11.1 | Design compliance with building construction technical requirements and criteria | 0.735 | | | | | 0.223 |

From these tests, which consist of outer model and inner model tests, several things can be known regarding the proposed model, including:

1. Model Validity

The first step in determining convergent validity is to examine the outer loading value. An indication is considered legitimate if its outer loading value is greater than 0.70 [54,60]. In the summary table, there is a loading factor value for each construct employed against the latent variable, and from there, it is determined that the loading factor values satisfy the criteria (all are above 0.7) and may be considered legitimate. In addition to outer loading, testing is also conducted using the extracted value of the average variance (AVE). Variables are considered legitimate if their AVE values are greater than 0.5 [54], and according to Table 6, all variables' AVE values are greater than 0.5; hence, they may be considered valid.

The previous stage consisted of the convergent validity test and the next step consists of the discriminant validity test. The first stage testing discriminant validity examined the Fornell–Larcker criterion. A test can be said to be valid by comparing the square root value of AVE for each variable with other variables having the highest value [54]. From the data processing table, it can be seen that all variables met the criteria for testing using the

Fornell–Larcker criterion. Apart from using the Fornell–Larcker value, testing discriminant validity can also be completed using the cross-loadings method. Convergent validity testing is performed by comparing the outer loading value of an indicator with its variable; the outer loading value must be greater than the indicator with other variables [54]. The data for the calculation of cross-loading are listed in the data processing table with the test results, which can be said to be valid, because the loading factor value for each indicator associated with the variable is greater than that of other variables. With all these tests carried out, it can be said that the model proposed in this study is considered valid.

To find out which indicators are most relevant besides using RII in explaining the latent variable, a confirmatory factor analysis (CFA) approach was used. CFA is part of SEM measurement, which shows the relationship between latent variables and their indicators [61]. Confirmatory Factor Analysis (CFA) is one part of the SEM (Structural Equation Modelling) method, which serves to test and analyze existing hypothesis relationships between indicators and latent variables [54]. This CFA method consisted of tests carried out on the measurement model, such as validity and reliability testing. In determining which indicators are most relevant, a factor loading approach was used, which is part of the validity testing of convergent validity. Factor loading is basically a coefficient that shows the relationship between indicators and their latent variables [61]. There are several methods used in testing this factor loading, and one of them was used to test it, and also to determine which indicators had a high influence by using the outer loading values. Outer loadings are estimates of relationships in reflective measurement models (i.e., arrows from latent variables to their indicators). They determine how high the influence of an indicator is for the assigned construct (latent variable) [54].

Based on method in Figure 6, indicators that have an outer loading value below 0.7 were eliminated; thus, we obtained relevant indicators for each variable, as listed in Table 6. This is the reason why not all variables that were compiled previously can be found in the SEM model that we produced.

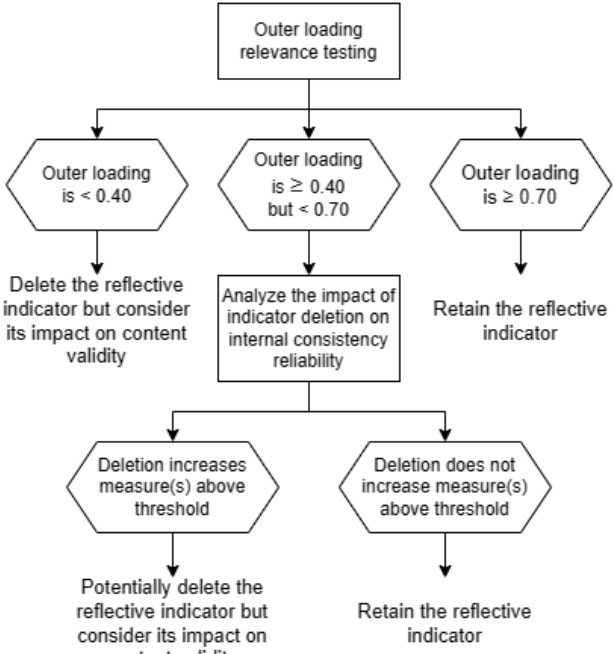

**Figure 6.** Testing the relevance of outer loading (Reprinted/adapted with permission from Ref. [62]. 2017. Hair et al.).

2. Model Reliability

After the validity test was carried out, a reliability test was then carried out to determine whether the model used was reliable or not. The first step that needed to be taken

was to conduct the composite reliability test, where a variable can be said to be reliable if the composite reliability value is above 0.7 [54]. Based on the test results, the data show that the composite reliability values are all above 0.7, so, according to this test, the data can be said to be reliable. Furthermore, the Cronbach's alpha test was carried out to determine its reliability as well. These test data can be said to be reliable if the Cronbach's alpha value is above 0.7 [54]. From the tests that were carried out, it was found that each latent variable used in this study has a Cronbach's alpha value above 0.7; so, the data and model can be considered reliable. By conducting these two tests, it can be said that the data and model for this study can be said to be reliable.

3.  Structural Test—$R^2$

After conducting validity and reliability tests included in the outer model test, the inner model test was then carried out. The inner model test was carried out with the aim of seeing whether the relationship between latent variables, namely exogenous and endogenous constructs, was able to provide answers to questions regarding the relationship between latent variables that were hypothesized previously. The first test conducted was the $R^2$ test. The $R^2$ value indicates how much influence variable X (exogenous construct) in the study has on variable Y (endogenous construct) [54]. From the test results, it was found that the $R^2$ values for variables Y1, Y2, and Y3 were 0.332, 0.596, and 0.567, respectively. For its own value, the $R^2$ value can be categorized as follows (Table 7):

**Table 7.** Category for $R^2$ value [54].

| $R^2$ Value | Category |
|:---:|:---:|
| <0.25 | Very Weak |
| $0.25 \leq R^2 < 0.50$ | Weak |
| $0.50 \leq R^2 < 0.75$ | Moderate |
| $0.75 \leq R^2$ | Substantial |

From the Table 7, it can be seen that the $R^2$ value for variable Y1 (Initiating Phase) falls into the weak category and that variables Y2 (planning Phase) and Y3 (construction phase) fall into the moderate category. This value shows that in the proposed model, the planning and design and construction phases in the implementation of sustainable construction using BIM were explained by 56.7% and 59.6%. The planning variable attained an R-Square value of 0.332, which, when compared to the literature, was classified into the weak category. The selected endogenous construct variables and their indicators can only explain the planning variable up to 33.2% and there is still as much as 62.8% attributable to other factors that need to be considered again in order to better explain the endogenous construct.

4.  Predictive Relevance Test—$Q^2$

The predictive relevance or $Q^2$ test is a test conducted to determine how good the observation value is by using the blindfolding procedure in the SMARTPLS® software by looking at the $Q^2$ value. If the value of $Q^2 > 0$, then it can be said to have a good observation value, whereas if the value of $Q^2 < 0$, then it can be stated that the observation value is not good [63]. From the test, it was found that the $Q^2$ value for each variable was greater than zero. This indicates that the proposed model has a good observation value.

5.  Model Fit Test

The Model Fit test is also one of the most frequently used non-parametric tests. The goal of the test is to determine how appropriate the observed frequency is to the expected frequency [64]. Lohmöller (1989) explained several approaches in testing model fit in SEM PLS. Some of them include standard root mean square (SRMR) and root mean square theta (RMS Theta). The following are the SRMR and RMS Theta results for the proposed model (Table 8):

**Table 8.** Model fit test results.

| Parameter | Value |
|---|---|
| SRMR | 0.083 |
| RMS Theta | 0.089 |

SRMR is defined as the difference between the observed correlation and the model implied correlation matrix, where this method (SRMR) was proposed by Henseler et al. (2014) to avoid conceptual errors. The model can be said to be valid if the SRMR value is smaller than 0.10 or 0.08 [65]. Meanwhile, RMS Theta is the root mean square residual covariance matrix of the outer model residuals [64]. RMS Theta values below 0.12 indicate a fit model, while higher values indicate a lack of fit model [66]. Based on these two requirements, the existing model results can be said to be model fit.

6. Path Coefficient

After testing and determining the data to be valid, there were several things that could be analyzed related to the SEM model. The first analysis was related to the relationship between exogenous construct variables (BIM attributes) to endogenous construct variables (sustainable construction implementation criteria). This can be seen from the path coefficient listed in the model image that has been made. In the path coefficient, there were several variables that had a negative value (−) on the endogenous construct variable, as shown in the following Table 9:

**Table 9.** Path coefficient value.

| Variable | Initiating Phase | Planning Phase | Construction Phase |
|---|---|---|---|
| BIM Design Function | 0.370 | −0.093 | −0.103 |
| BIM Build Function | −0.195 | 0.146 | 0.043 |
| BIM Operate Function | −0.020 | −0.110 | 0.065 |
| SC Barrier | 0.044 | −0.070 | −0.041 |
| BIM Barrier | −0.021 | 0.041 | −0.035 |
| SC Driver | 0.440 | 0.593 | 0.672 |
| BIM Driver | −0.097 | 0.207 | 0.070 |
| Awareness Based Strategy | 0.064 | 0.086 | 0.142 |
| Regulation Based Strategy | 0.242 | −0.106 | −0.062 |
| Company Based Strategy | −0.193 | 0.081 | −0.025 |

The negative value of the path coefficient indicates a negative influence on the path relationship, so we can determine which BIM attributes from each category are most appropriate for each stage in the implementation of sustainable construction (construction, programming, and planning) by looking at which path coefficient is positive from the Table 9.

3.2.2. Results of Relative Importance Index Testing

The next step was to review which factors are included in the barriers and benefits contained in the practice of applying BIM technology to sustainable construction. A ranking was completed using the Relative Importance Index (RII) approach. The RII value was obtained from the division between the weighting of the sum of all respondents' answers (W) divided by the highest weight (A = 5) and the number of respondents (N = 60). From there, the following Table 10 was obtained, which contains the most relevant indicators for each variable based on the RII calculation along with the RII value:

**Table 10.** Most-relevant indicators for each variable based on RII value.

| Code | Variable | Code | Indicator | RII |
|---|---|---|---|---|
| X1.1 | BIM Design Function | X1.1.1 | Building digitization | 0.880 |
| X1.2 | BIM Build Function | X1.2.4 | Further development of 2D CAD | 0.923 |
| X1.3 | BIM Operate Function | X1.3.4 | Integration between different tools for each party | 0.883 |
| X2.1 | SC Barrier | X2.1.2 | Cost of implementation construction that tend to be high | 0.830 |
| X2.2 | BIM Barrier | X2.2.2 | BIM implementation requires higher initial investment than conventional methods | 0.857 |
| X3.1 | SC Driver | X3.1.4 | Reduce construction waste | 0.847 |
| X3.2 | BIM Driver | X3.2.8 | Improve visualization for stakeholders | 0.913 |
| X4.1 | Awareness Strategy | X.4.1.3 | Conduct more research in the use of BIM for sustainable construction | 0.903 |
| X4.2 | Regulation Strategy | X4.2.2 | Create encouragement, support, and commitment from regulators to accommodate the implementation of BIM for sustainable construction | 0.930 |
| X4.3 | Company Strategy | X4.3.2 | Prepare infrastructure (tools, software, hardware, etc.) to be able to apply BIM into sustainable construction | 0.890 |

After collecting the results of data processing using SEM, processing proceeded by linking any BIM variables in each category that had a connection with each of the established sustainable construction criteria by determining whether the path coefficient value was positive. In addition, we wished to determine which relevant indicators would be used to represent these variables in the tabulation results by picking the RII processing results with the greatest value for inclusion in the relevant indicators for the variable.

### 3.3. Research Findings

Based on the modelling carried out using the SmartPLS®, a software for data analysis using the PLS-SEM method by a company named SmartPLS GmBH from Germany, data related to the modelling regarding the relationship between the attributes of the BIM and the criteria for implementing sustainable construction are known. From the model, one can observe the relationship of the BIM variable to the endogenous construct variable (sustainable construction implementation criteria). This can be seen from the path coefficient in the figure related to the SEM model used. The following are the conclusions regarding which BIM indicators are relevant in representing the relationship between BIM and sustainable construction based on the results of SEM analysis (Table 11):

**Table 11.** Research Findings—correlation between BIM and sustainable construction.

| Variable X (BIM) | Relevant Indicator | Its Effect on Variable Y |
|---|---|---|
| (BIM) | (Based on RII) | (Sustainable Construction Criteria) |
| Design Function | Building digitization | Initiating Phase |
| Build Function | Further development of 2D CAD | Planning Phase<br>Construction phase |
| Operate Function | Integration between different tools for each party | Construction phase |
| SC Barrier | Cost of implementation construction tend to be high | Initiating Phase |
| BIM Barrier | BIM implementation requires higher initial investment than conventional methods | Planning Phase |
| SC Driver | Reduce construction waste | Initiating Phase<br>Planning Phase<br>Construction phase |
| BIM Driver | Improve visualization for stakeholders | Planning Phase<br>Construction phase |
| Awareness Strategy | Conduct more research in the use of BIM for sustainable construction | Initiating Phase<br>Planning Phase<br>Construction phase |
| Regulation Strategy | Create encouragement, support, and commitment from regulators to accommodate the implementation of BIM for sustainable construction | Initiating Phase |
| Company Strategy | Prepare infrastructure (tools, software, hardware, etc.) to be able to apply BIM into sustainable construction | Planning Phase |

## 4. Discussion

This study shows how to implement BIM into sustainable construction workflows using a systematic approach. From the research findings, we can determine, for the four major BIM variables, such as BIM functions, barriers, drivers, and implementation strategies, which indicators are considered most important and at what phase of sustainable construction they should be implemented. Each phase of sustainable construction consists of a list of activities that must be accomplished during that phase of construction in order for the project to be called a sustainable construction project. Despite the information provided above, how does this relationship model operate? To answer this, we could use the variable "BIM function" in the design phase (X1.1) as an explanation. When we are talking about BIM function in the design phase, we examine Table 6: a summary of SEM model data testing. This table contains four relevant indicators for the variable "BIM function" in the design phase (X1.1) when there were initially five. These indicators, namely building digitization, integration between parties involved, feasibility study, and energy efficiency analysis, are the relevant variables when it comes to BIM implementation for sustainable construction in Indonesia. This method might be implemented in other similar cases, such as developing countries. However, further research still needs to be done to validate this claim.

Moreover, when we are talking about sustainable construction criteria, this research divides the topic into 3 main phases, namely the initiating, planning, and construction phases, which are derived from Indonesia's Ministry of Public Works and Housing Regulation No. 9 of 2021, where this regulation discusses how the construction sector in Indonesia should implement sustainable construction into their construction projects. Each variable consists of its own respective indicators. However, they are mainly focused on the same thing: requirements that the construction sector in Indonesia must fulfill. Despite all this, this

regulation did a good job of implementing the concepts of sustainability and sustainable construction into these requirements for each phase of the three phases of sustainable construction projects. The notion of preserving resources for future generations and the three pillars of sustainability were well-addressed in this regulation. From the results of this research, we can examine the "initiating phase" variable as an example. This variable has correlating indicators for the three pillars of sustainability, such as: (a) Availability of development plans that are responsive to gender, people with disabilities, and marginalized people (Y.1.5.1) that fulfill the social aspect; (b) Availability of development plans that support regional economic development (Y.1.6.1) that fulfill the economic aspect; and (c) Availability of development plans in accordance with technical standards and utilization of environmentally friendly technology (Y.1.7.1) that fulfill the environment aspect.

Moreover, this indicator has the greatest influence on the initiating phase of the sustainable construction criteria, meaning that "building digitization" helps in the process of preparing for the availability of every development plan required based on this regulation. This process also applies to the other findings in Table 11 as well. In this way, BIM technology can be used to implement sustainable construction as well as contribute directly and indirectly to the social, economic, and environmental aspects of sustainability. Thus, based on the previous explanations and the findings of this study, we can conclude that the variable "BIM function" in the design phase (X1.1) has "building digitization" as its most relevant indicator. While this research was conducted to investigate how we implement BIM for sustainable construction using the function, barrier, driver, and strategy approaches, there are already many papers discussing the contribution of BIM technology to each aspect of sustainability, namely the environmental, social, and economic aspects.

First, we could examine sustainable construction from an economic perspective. There are several indicators used in this research that correlate with this perspective, such as "reduce operation and maintenance costs after construction". This aligns with the research that was conducted by Lei Zhou and David J. Lowe in 2003 about the economic challenge of sustainable construction. They found that the most significant economic benefits of sustainable design are reduced operation and utility expenses, reduced maintenance costs, and an overall improvement in the performance and efficiency of the building [67]. However, when we are talking about the economic perspective regarding BIM drivers, there were no relevant variables found, both in the SEM and RII analyses. The absence of relevant variables aligns with the research conducted by Haron et al. in 2017, wherein they stated that the implementation of BIM might reduce costs in developed countries, but it may not do so in developing countries [68], and even its implementation itself might be considered an additional cost for projects, especially considering that Indonesia, the study case for this research, is still considered a developing country. This fact also strengthens results of the analysis category "BIM barrier from the economic perspective", which shows that BIM implementation requires higher initial investment than conventional methods. In addition to the economic perspective, there is the social perspective. Some of the valid variables within this perspective include improving health and safety and availability of development plans that are responsive to gender, people with disabilities, and marginalized people. With BIM's ability to analyze and simulate a wide variety of variables, which, with traditional tools, would be extremely complicated and require manual data entry, complex analyses can be performed to improve working and living conditions, thus further enhancing comfort and well-being for every individual in the workforce [24]. Lastly, we can consider the environmental perspective. The environmental perspective is the pillar that has the most indicators in this study compared to the other two. However, this fact is not difficult to understand. The core of the environmental perspective revolves around nature and resource conservation. Regarding the environmental perspective, BIM facilitates a variety of analyses. Integration with other specialized tools, such as life cycle assessment (LCA), can improve its ability to maximize environmental performance [24].

## 5. Conclusions

As the construction sector has been significantly growing around the globe, it has begun leading to several major consequences that we should not neglect. These considerations include its environmental impact, which consumes approximately 42 percent of energy, 30 percent of raw materials, and 25 percent of clean water globally, as well as the sector itself, frequently referred to as an "essential economic engine" for its contribution to one-tenth of the global economy These complex considerations ultimately concern creating buildings and infrastructure for the well-being of the world's citizens while improving their quality-of-life, social interaction, and general well-being. However, not everyone is privileged enough to experience these quality-of-life enhancements. All these issues have been of concern for many years, yet this does not imply that the considerable growth of the building industry always leads to its negative consequences. Many studies have been conducted for decades to improve the environmental, economic, and social conditions of the building industry. In order to preserve our future in relation to these issues, the concept of sustainable construction has arisen as a suitable solution. Its capacity to construct while preserving three important aspects of our daily lives, environmental, social, and economic, has been viewed as an exact solution for humans, both now and in the future. Speaking of the future, regarding the future of the building industry, we have begun integrating information and communications technology (ICT) into the construction workflow. The conventional workflow is becoming outdated, and the "culture" is gradually transitioning to a fresher and more modern approach with several new possibilities on the horizon. Building information modelling (BIM) is the solitary example of this case in which practitioners and academics have expressed great concern. Its ability to create and manage information throughout a construction project's entire life cycle has resulted in many improvements. This research was derived from these two topics: sustainable construction and building information modelling (BIM). It arose from the question: "what would happen if we tried to implement BIM into sustainable construction?", and, finally, it has found its core idea: determining the relationship between each BIM factor and sustainable construction criteria. This research focused on functions, barriers, drivers, and implementation strategies, and was conducted by analyzing relationships among these using structural equation modelling (SEM). It was found that the BIM function has a positive influence on sustainable construction with relevant indicators in the form of building digitization, improvement from 2D CAD methods, and integration between tools. Relevant barriers include a lack of demand from clients and implementation that feels like additional work. Relevant drivers include increasing work productivity and reducing work errors. Meanwhile, relevant strategies include conducting further research, providing commitment, and setting up infrastructure for the application of BIM into sustainable construction. In the subcategory of sustainable construction criteria, in this research, initiating, planning, and construction phases served as relevant indicators occupying their own specific positions. It is clear that each of the above-mentioned relevant variables can contribute to and improve the performance of sustainable construction activities in their respective relationships. Thus, BIM technology can be used to implement sustainable construction practices as well as contribute directly and indirectly to the social, economic, and environmental aspects of sustainability. Future research on this topic should concentrate more on the performance of this approach and whether it is still applicable to the construction industry outside of Indonesia, knowing that this research is solely an approach based on a respondent questionnaire and that no testing has been performed. In addition, this research on the integration of BIM into sustainable construction does not prevent the integration of other available ICTs.

**Author Contributions:** Conceptualization, C.K.M.; Methodology, C.K.M.; Validation, F.M.; Formal analysis, C.K.M.; Investigation, C.K.M.; Resources, C.K.M.; Data curation, F.M.; Writing—original draft, C.K.M.; Writing—review & editing, C.K.M.; Visualization, C.K.M.; Supervision, F.M.; Project administration, F.M.; Funding acquisition, F.M. All authors have read and agreed to the published version of the manuscript.

**Funding:** This research received no external funding.

**Institutional Review Board Statement:** Not applicable.

**Informed Consent Statement:** Informed consent was obtained from all subjects involved in the study.

**Data Availability Statement:** Not applicable.

**Conflicts of Interest:** The authors declare no conflict of interest.

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
