# Peer review of "Relationship between Functions, Drivers, Barriers, and Strategies of Building Information Modelling (BIM) and Sustainable Construction Criteria: Indonesia Construction Industry"

_sustainability, doi:10.3390/su15065526_

Round 1
Reviewer 1 Report
Please see the document attached for detailed comments.

Author Response
Thank you for your review; for my answer to your review, Please refer to the file attached below.

Reviewer 2 Report
Comment 1:
L87-L96: The list of benefits associated to the use of BIM could expand, for example to cover LCC, quality management, quantity take-off, and risk management.
Comment 2:
L120-L121: Could not find details of Abdullah Al-Yami in 2019 in the list of references.
Comment 3:
L144-L147: Could not find details related to The Minister of Public Works and Housing released Ministry of Public Works and Housing Regulation Number 9 of 2021.
Comment 4:
L235: Need to clarify if the four research questions are the same than the four key variables cited in L238. It would be good if these research questions are presented explicitly and briefly discussed.
Comment 5:
L366: Figure 2 is not of good quality.
Comment 6:
L468: The font size in Figure 6 is too small to be read.
Comment 7:
L518: Figure 7 is not of good quality.
Comment 8:
L699: The list of bullet points does not demonstrate critical thinking while leaving aside constraints encounter in the study and suggested strategies to maximise benefits in the Indonesian market derived from the use of BIM.

Author Response

(The authors gave the same response as above.)

Reviewer 3 Report
- What is the main question addressed by the research?
This paper presents Relationship between Functions, Drivers, Barriers, and Strategies of Building Information Modelling (BIM) to Sustainable Construction Criteria: Indonesia Construction Industry. The main focus of this work is functions, barriers, drivers, and implementation strategies, which will be analyzed for their relationship with sustainable construction criteria, which is done by analyzing the relationship using structural equation modelling (SEM) and this can be interesting and suitable for the Sustainability, MDPI.
2. Do you consider the topic original or relevant in the field? Does it
address a specific gap in the field?
Yes, it is relevant and it can adress a specific gap in the field. The authors asked good questions and supported with the 55 articles related with this field.
3. What does it add to the subject area compared with other published
material?
The authors stated the research questions which could be important additions to the research area. The questions are given below. They tried to answer in scientific manner.
Analyzing the relationship between BIM functions and sustainable construction criteria;
• Analyzing the relationship between barrier factors in the application of BIM in sustainable construction and sustainable construction criteria;
• Analyzing the relationship between the driving factors for the application of BIM in sustainable construction and sustainable construction criteria;
• Analyzing the relationship between strategies for improving the application of BIM in sustainable construction and sustainable construction criteria.
4. What specific improvements should the authors consider regarding the
methodology? What further controls should be considered?
Research Analysis Method is explained well with sub sections. I think this is appropriate.
5. Are the conclusions consistent with the evidence and arguments presented
and do they address the main question posed?
The conclusion part is a bit general and let there be sentences with specific results.
6. Are the references appropriate?
The references used in the study are also within the scope of the study. However, the reference format must be adapted. There are redundant statements such as available online among the references. These should be corrected.
7. Please include any additional comments on the tables and figures.
The seven Figures and eleven Tables are used in this study. The some Figures used in the article are not of good quality and readable. Please, try to improve these Figures. In Tables, some parts have been started with lower case letters, make these in upper case.
Author Response

(The authors gave the same response as above.)
